# Recognition of Idiopathic Inflammatory Myopathies Underlying Interstitial Lung Diseases

**DOI:** 10.3390/diagnostics15030275

**Published:** 2025-01-24

**Authors:** Giulia Morina, Domenico Sambataro, Alessandro Libra, Stefano Palmucci, Michele Colaci, Gaetano La Rocca, Francesco Ferro, Linda Carli, Chiara Baldini, Santa Valentina Liuzzo, Carlo Vancheri, Gianluca Sambataro

**Affiliations:** 1Department of Clinical and Experimental Medicine, Regional Referral Center for Rare Lung Diseases, Policlinico “G.Rodolico-San Marco”, University of Catania, 95123 Catania, Italy; giulia.morina@tiscali.it (G.M.); alessandrolibra@outlook.it (A.L.); valeliu92@gmail.com (S.V.L.); vancheri@unict.it (C.V.); 2Artroreuma S.R.L., Outpatient Clinic Associated with the Regional Health System, Mascalucia, 95030 Catania, Italy; d.sambataro@hotmail.it; 3Unità Operativa Semplice Dipartimentale di Imaging Polmonare e Tecniche Radiologiche Avanzate (UOSD IPTRA), Department of Medical Surgical Sciences and Advanced Technologies “GF Ingrassia”, University Hospital Policlinico “G.Rodolico-San Marco”, University of Catania, 95123 Catania, Italy; spalmucci@unict.it; 4Internal Medicine Unit, Division of Rheumatology, Department of Clinical and Experimental Medicine, Cannizzaro Hospital, University of Catania, 95123 Catania, Italy; michele.colaci@unict.it; 5Rheumatology Unit, Department of Clinical and Experimental Medicine, University of Pisa, 56126 Pisa, Italy; gaelarocca94@gmail.com (G.L.R.); francescoferrodoc@gmail.com (F.F.); 81clinda@gmail.com (L.C.); chiara.baldini74@gmail.com (C.B.); 6Department of Medicine and Surgery, University of Enna “Kore”, 94100 Enna, Italy

**Keywords:** interstitial lung disease, idiopathic inflammatory myopathies, dermatomyositis, polymyositis, antisynthetase syndrome, autoantibodies

## Abstract

Interstitial Lung Disease (ILD) is one of the most common causes of mortality in idiopathic Inflammatory Myopathies (IIM). Despite these conditions being commonly associated with proximal weakness, skin rashes and arthritis, ILD can be the first or the sole clinical feature in up to 60% of patients, potentially leading to incorrect diagnosis. The early recognition of an underlying IIM in ILD patients can allow for prompt treatment, which could potentially stabilize or even improve the lung disease, also avoiding the development of other clinical features associated with the condition. The objective of this review is to describe the clinical, serological and radiological features associated with IIM-ILD, mainly focusing on dermatomyositis and antisynthetase syndrome.

## 1. Introduction

The term “Interstitial Lung Disease” (ILD) defines the deposition of cells and/or extracellular matrix in the lung interstitium. While in some cases, the disease may regress or stabilize, in other cases, especially in the presence of a fibrotic phenotype, it can progress, leading to decreased quality of life or even death due to respiratory failure [1].

Over 200 different conditions are associated with ILD, such as environmental exposure (smoke, dust, drugs), genetic factors, infections and autoimmune conditions. Among the latter group, Connective Tissue Diseases (CTDs) are responsible for about 25% of ILDs [2]. In particular, about 6% of the total ILD is caused by a subgroup of CTDs called Idiopathic Inflammatory Myopathies (IIM) [3].

The diagnosis of an IIM underlying ILD is not straightforward because it requires tight collaboration between rheumatologists and pulmonologists for the recognition of clinical and serological signs, which are often nuanced. Moreover, as seen in primary Sjögren’s Syndrome (pSS), ILD may be the first or even the sole clinical manifestation of IIM reported by patients in 20–60% of cases [4,5,6,7]. Those patients are commonly classified as idiopathic, especially in the presence of fibrotic features identified by High-Resolution Computed Tomography (HRCT) or biopsy [8,9].

Despite ILD being a common manifestation of IIM, it was not included in the classification criteria for IIM or DM [10,11,12], which are currently the most widely used criteria for the classification of these patients, contributing further to the diagnostic delay.

Clearly, an incorrect diagnosis may delay an appropriate therapeutic approach, which may have a potential impact on survival.

The objective of this review is to describe the diagnostic approach to ILD patients, with a focus on the identification of potentially underlying IIM.

## 2. Classification of Idiopathic Inflammatory Myopathies

The term IIM encompasses a group of CTDs characterized by proximal weakness and prevalent inflammation of muscles. The group initially included only Polymyositis (PM) and Dermatomyositis (DM), according to the criteria proposed by Bohan and Peter in 1975 [10,11]. In the following years, Inclusion Body Myositis (IBM), Immune-Mediated necrotizing Myopathy (IMNM) and Antisynthetase Syndrome (ASyS) were added [12,13,14], and some authors suggested including Overlap myositis [15], conditions in which patients have myositis associated with another CTD. Increased knowledge of myositis histology, along with the discovery of a number of autoantibodies serving as disease markers, has given rise to suggestions of removing PM from this classification, as most patients proved to be more correctly classified as IBM, IMNM, or ASyS [14].

At present, the sole classification criteria available for the recognition of these conditions are the historical Bohan & Peter Criteria [10,11], while ASyS is classified according to expert opinion-based criteria [16], pending the forthcoming criteria endorsed by the American College of Rheumatology (ACR) and the European Alliance for Associations for Rheumatology (EULAR) [17].

As DM and ASyS are responsible for the vast majority of IIM-ILD, this review mainly focuses on these two entities.

## 3. Clinical Features of Dermatomyositis and Antisynthetase Syndrome

The clinical manifestations of DM and ASyS are mainly associated with skin and muscle involvement. These features can be very useful for diagnosis. However, it should be noted that up to 40% of non-Jo1 + ASyS patients will only show ILD during their entire clinical course [18]. A comprehensive list of possible clinical features is reported below.

### 3.1. Unexplained Fever

Fever is quite common in IIM, with a prevalence ranging from 19–41% [17,19]. This symptom seems to be more common in DM than in ASyS, probably due to its underlying pathogenic mechanism driven by interferon 1 [20]. Generally, it is described as an increased oral body temperature of 37.5 °C for 3 days [19]. Significantly higher values can help differentiate an autoimmune origin from an infectious one, although this is not always straightforward. Some infections, such as COVID-19 pneumonia, can induce an inflammatory ILD very similar to that expected in IIM-ILD and can even be associated with the production of anti-MDA5 autoantibodies [21,22]. Of note is the fact that the presence of fever is mainly associated with the presence of consolidations on HRCT and an increased risk of acute exacerbation [23].

### 3.2. Raynaud’s Phenomenon

Raynaud’s Phenomenon (RP) is a relatively common sign in both healthy individuals and CTD patients. RP results from the vasoconstriction of terminal circulation (mainly in the hands, but also feet, ears and nose), with a color change from white to blue and red (at least the first two phases must be present to define the condition) [24]. The prevalence of RP is 3–5% in healthy subjects; it is almost always present in Systemic Sclerosis (SSc), while it affects about 20–30% of IIM patients [25,26,27]. Nailfold Videocapillaroscopy (NVC) is a very useful tool to determine whether RP is isolated (“primary RP”) or associated with an underlying autoimmune disease: in the presence of SSc, NVC findings include giant capillaries and/or avascular areas [28]. NVC can also be pathologic in IIM, with the possible presence of bushy capillaries [29]. However, differently from SSc, NVC positivity is not associated with the presence of RP in IIM patients [30,31]. Therefore, it could be useful to perform NVC on all ILD patients for whom IIM is suspected.

### 3.3. Inflammatory Arthritis

Inflammatory arthritis, together with myositis and ILD, is part of the classic triad used to define the typical clinical pattern of ASyS [32]. Prevalence ranges from about 15% in DM patients to 45% in ASyS patients [17]. Different patterns of arthritis have been described: if present at the onset of ASyS (about 30% of cases), arthritis shows a distribution and clinical behavior similar to Rheumatoid Arthritis (RA), with symmetrical involvement of the small joints and the potential presence of erosions. If inflammatory arthritis develops during the disease course (about 15% of cases), it resembles a more common arthritis seen in CTDs (asymmetrical, without erosions) [33].

### 3.4. Myositis

Myositis is present in up to 70% of ASyS cases during the clinical course, and it is virtually always present in DM, although some DM cases, mainly those associated with anti-MDA5 positivity, can present a clinically amyopathic phenotype [17,34]. The presence of myositis may be suggested by increased levels of muscle enzymes: Lactic Dehydrogenase (LDH), Aspartate and Alanine Transferase (AST and ALT), and other more specific markers such as Creatine Phosphokinase (CPK), myoglobin and aldolase [35]. The most important enzyme for diagnosing myositis is CPK, although aldolase may be increased with normal CPK levels, suggesting early muscle cell regeneration [36]. Clinically, myositis typically involves the proximal muscles of the arms and legs, as well as neck flexors, pharynx and the proximal esophagus, leading to fatigue, proximal weakness, and, more rarely, dysphagia [37]. These two latter symptoms are considered very important for a correct diagnosis and were included in the classification criteria for IIM [12]. The clinical picture often also includes myalgia, reported by about 40% of patients [17]. While myalgia is common, it is deemed nonspecific and was not included in the classification criteria for any IIM. However, it could be diagnostically valuable, especially in patients presenting with ILD. ILD-IIM patients often show little or no increase in muscle enzymes, making diagnosis difficult [38]. However, ILD patients with myalgia have been found to test positive for myositis autoantibodies in up to 68% of cases [39].

Myositis can be confirmed by electromyography (EMG), commonly showing a myopathic pattern, or through imaging: ultrasound can suggest the presence of edema, while Magnetic Resonance Imaging (MRI) can provide more specific details on muscle inflammation and damage. Finally, Positron Emission Tomography (PET) can be useful both in evaluating muscle inflammation and as a potential screening tool for associated cancer [40,41]. Indeed, IIM patients present an increased risk of various cancers within 3 years of diagnosis [42].

Rarely, confirmation of myositis may require histological examination. DM is histologically characterized by CD4+ cells in the endomysium, B lymphocytes in perivascular areas, endothelial hyperplasia with thrombi, reduced capillary density, and perifascicular atrophy. In AsyS, macrophages predominate, with a similar distribution of CD4+ cells, B cells in the perimysium, and CD8 cells in the peri and endomysium. Perifascicular regions typically show necrosis [43,44].

### 3.5. Skin Involvement

Skin involvement is very important for diagnosing ILD-IIM. ASyS is typically associated with hyperkeratotic rashes, while DM is associated with erythematous lesions. Among the hyperkeratotic rashes, Mechanic’s Hands and Hiker’s Feet are notable. They are characterized by cracking and hyperkeratosis of the fingers (generally the first three) and the palms and plantar soles, respectively [45]. Regarding erythematous rashes, the most important are Gottron Papules. They are red-to-violaceous papules on the extensor surfaces of joints, primarily the hands. The lesion is considered pathognomonic for DM [46]. Another nearly pathognomonic feature is the Heliotrope Rash, a violaceous, dusky rash around the eyes, with or without associated edema. Other erythematous-violaceus rashes and macules may appear on the elbows and knees (Gottron Sign), the posterior neck and shoulders (Shawl Sign), the anterior neck and chest (V sign) and the thighs (Holster Sign) [46]. DM can also be associated with facial erythema involving the nasolabial folds and calcinosis [46].

It is important to highlight that, when facing a patient with ILD, the presence of florid erythematous skin involvement, especially if not associated with significant myositis, should raise a suspicion of Amyopathic DM, a condition generally associated with positivity for anti-MDA5 antibodies and a severe risk of rapid progression [47]. A collection of typical skin rashes associated with IIM is reported in Figure 1 and Figure 2.

## 4. Autoantibodies in Dermatomyositis and Antisynthetase Syndrome

Autoantibodies play a crucial role in diagnosing IIM in the context of ILD potentially associated with IIM. In recent years, many new myositis-related autoantibodies have been described. The majority of these are associated with the presence of ILD, and each autoantibody typically correlates with a specific clinical presentation [48].

### 4.1. Types and Classification of Myositis Autoantibodies

Myositis autoantibodies are commonly divided into two categories, namely Myositis Associated Autoantibodies (MAA) and Myositis Specific Autoantibodies (MSA). The former includes anti-PM/Scl, anti-Ku, anti-U1RNP and anti-Ro52kD. These autoantibodies are often present in overlap conditions or may also be found in CTDs other than IIM. In contrast, MSA is regarded as highly specific for different IIM subtypes [49]. Table 1 summarizes the main IIM-related autoantibodies, categorizing them according to the specific disease subtype with which they are associated [50].

Considering MAA from a pneumological perspective, the most important autoantibodies are probably anti-Ro52kD. They may be detected in virtually all CTD subtypes, but they are associated with Antisynthetase Antibodies (ASA) in up to 50% of cases of ASyS, portending a worse prognosis in ILD patients [54]. In clinical practice, it is not uncommon to find anti-Ro52kD autoantibodies in the sera of ILD patients in the absence of any extrapulmonary sign of CTDs. These patients are usually classified as “Interstitial Pneumonia with Autoimmune Features” (IPAF) [66]. In these cases, a thorough assessment of the autoimmune profile, including IIM-related autoantibody testing, is warranted to detect potentially associated ASA. Moreover, IPAF patients should undergo a prompt follow-up to promote early recognition of potential progression toward specific CTDs [67,68].

The remaining MAA are anti-Pm/Scl, anti-Ku and anti-U1RNP. The first two are associated with an overlap condition between SSc and myositis called “scleromyositis”, while anti-U1RNP is the typical marker of Mixed Connective Tissue Disease (MCTD), a rare condition characterized by an overlap between myositis, Systemic Lupus Erythematosus and SSc. ILD is generally present in scleromyositis and MCTD in an established clinical picture; therefore, the diagnosis is generally quite straightforward.

ILD secondary to IIM (ILD-IIM) is closely associated with MSA, mainly in the subclass of Dermatomyositis Antibodies (DMA) and ASA. DMA include anti-MJ/NXP2, anti- Tif1γ, anti-Mi2, anti-MDA5 and anti-SAE. All of these autoantibodies are associated with significant erythematous skin involvement. Myositis is generally more severe in anti-MJ/NXP2, whereas anti-MDA5 is associated with a severe interferon-1-mediated interstitial lung disease, potentially rapidly progressive with a poor prognosis [34]. The frequent associations of DM with cancer should also be highlighted. Of note, cancer-related DM is generally associated with the presence of anti- Tif1γ autoantibodies [59].

ASAs are a group of MSAs representing the serological hallmark of ASyS. The classic triad of inflammatory arthritis, ILD, and myositis is more common with anti-Jo1 positivity (which is also the most common ASA) [32]. Other autoantibodies are rarer but are associated with severe ILD, which can be the sole ASyS manifestation during the entire course of the disease [18]. Since anti-Jo1 is the only ASA detected by commonly available commercial kits for Extractable Nuclear Antigen, specific testing for non-jo1-ASA should be performed whenever the suspicion of secondary ILD is raised [68].

### 4.2. Interpretation and Reliability of Myositis Autoantibodies

Correct interpretation of MSA/MAA is crucial in the management of ILD patients. Therefore, it is important to highlight current limitations in autoantibody testing. The current gold standard for the assessment of MSA/MAA is Immunoprecipitation (IP), which is an expensive, time-consuming technique and, therefore, not available on a large scale. Moreover, incorrect results are possible even with IP testing due to the potential co-migration of other proteins affecting the recognition of some autoantibodies (e.g., anti-MDA5, anti-NXP2, anti-SRP, anti-SAE) [48]. A certain variability has been shown in IP performed in different centers [69]. Other techniques, such as Line Blot Immunoassay (LIA), can recognize different autoantibodies at the same time in a semi-quantitative manner. They are easy and cheap to perform, but they are also burdened by a significant number of incorrect results. It is relatively common to see false positivity for MSAs such as anti-TIF1γ, anti-MDA5, anti-Mi2, and false negativity for rare ASA [70,71,72]. The optimal approach could be a periodic test in selected patients, followed by confirmation through IP in patients who test positive on LIA for MSA/MAA. However, other measures should also be employed to improve the reliability of LIA. First of all, MSA/MAA testing shows low reliability in populations with a low pre-test probability of IIM diagnosis [72]. Correlation with the expected clinical picture is useful in the general population [73] but probably less so with ILD patients since isolated ILD is very common in MSA-positive patients.

Correlation with the expected Antinuclear Antibody (ANA) pattern has also been shown to improve the reliability of LIA results [74]. The expected ANA pattern is reported for each autoantibody in Table 1. Moreover, false positivity is more common for low autoantibody titers and rare in the presence of high titers [75]. Inconclusive results, such as the “grey zone” in LIA, should be treated as negative. Finally, MSAs are considered mutually exclusive [49]: the association of multiple MSAs and the association of anti-Pm/scl with MSA is extremely uncommon [76]. However, some authors reported the coexistence of multiple autoantibodies, repeatedly confirmed and correctly associated with the clinical picture in about 10% of patients [77,78]. This is in line with clinical practice, where it is relatively common to find patients with seropositive ASyS showing clinical features associated with other conditions (mainly SSc). Therefore, positivity for multiple autoantibodies should be treated with caution by carefully considering the overall clinical picture, autoantibody titers and the expected ANA pattern. Importantly, the high risk of false positivity and the possibility of a potential overlap syndrome should be considered.

## 5. ILD Associated with Dermatomyositis and Antisynthetase Syndrome

ILD is a major cause of mortality in IIM, with an increased mortality rate of around 40% [79]. Respiratory failure is pathologically linked with the infiltration of inflammatory cells and/or the deposition of extracellular matrix in the lung interstitium. However, weakness of the respiratory muscles can also contribute [80]. As pointed out before, ILD can be the main or even the sole clinical feature in the context of IIM. Therefore, recognition of the underlying autoimmune disease can be difficult. Of note, patients with isolated lung involvement tend to be referred to lung units for respiratory symptoms prior to any rheumatologic assessment [17,81].

It should be noted that from the clinical point of view, respiratory symptoms of IIM-ILD patients are the same as those reported for other ILD patients (dyspnea, dry cough), but Velcro crackles on chest auscultation are uncommon, given the rarity of a UIP pattern in these conditions [3]. Attention should be given to the possibility that a significant proportion could be asymptomatic, as well as the severity of ILD in IIM and the myositis disease activity [3]. In addition, Pulmonary Function Tests (PFTs) can be sensitive but not specific. They can be normal or show a restrictive pattern characterized by a reduction in forced vital capacity and diffuse lung capacity for carbon monoxide. It is of great interest that a restrictive pattern can also be due to the inflammatory involvement of respiratory muscles rather than ILD. In these cases, the evaluation of maximal inspiratory and respiratory pressure (respectively, Total Lung Capacity and Residual Volume) has been proposed [82]. In any case, clinics, objective exams, and PFTs can be used as a first diagnostic assessment, but appropriate imaging is required. Their role had greater value during follow-up.

Chest X-ray can recognize parenchymal abnormalities, but usually when present at advanced stages [83]. HRCT is, therefore, the gold standard for detecting ILD, monitoring the response to treatment and identifying disease progression and/or complications. Importantly, HRCT allows early diagnosis of ILD and is useful to localize pulmonary lesions, estimating their extent and assess the fibrotic burden. HRCT is also crucial for prognostic purposes, as it can highlight the presence of ventilation defects and aspiration pneumonia. However, radiological findings alone are not sufficient for the proper diagnosis and classification of ILD patients; therefore, a multidisciplinary approach is warranted.

### 5.1. Radiological Pattern of ILD in IIM

The most common ILD pattern associated with the presence of IIM is Nonspecific Interstitial Pneumonia (NSIP), with or without superimposed Organizing Pneumonia (OP) (Figure 3). The NSIP pattern can be divided into cellular and fibrotic. Both are characterized by the presence of bibasilar and symmetric ground-glass opacities, with the fibrotic form adding fine reticulations. OP is characterized by ground-glass consolidations and rarely by reticulation, bronchiectasis, interstitial nodules, interlobular thickening of septa, halo sign, inverted halo sign or airspace nodules. OP lesions have a subpleural/peribronchovascular and perilobular distribution, with bilateral involvement, especially in the middle-basal lung regions [84]. These two patterns are present in about 66% and 32% of DM patients, respectively. However, when considering only Rapidly Progressive ILD-DM cases, the proportion of NSIP and OP patterns shifts to 30% and 70%, respectively, clearly suggesting that consolidations should be considered a marker of potential rapid progression, commonly associated with the presence of anti-MDA5 antibodies [85]. These autoantibodies are also associated with a possible acute Diffuse Alveolar Damage (DAD) pattern [85,86]. This pattern, although rare, is very similar to that observed in COVID-19 pneumonia, potentially associated with pneumothorax and/or subcutaneous emphysema [87]. The Usual Interstitial Pneumonia (UIP) pattern is also uncommon in DM-ILD, being present in about 2% of patients [86]. It is characterized by traction bronchiectasis, honeycombing and thickening of bronchovascular bundles [86].

ILD is also the main clinical sign of ASyS, affecting about 80% of patients in the largest cohort of ASYS available in the literature [17]. The most commonly encountered pattern, characterizing about 60% of cases, is also NSIP with or without associated OP. However, a UIP pattern is also possible, with a prevalence of about 12% of patients [7]. Mainly in the presence of non-Jo1 ASyS antibodies, ILD can be the first, or even the only, clinical feature associated with the condition. Notably, the prognosis seems to be more severe in the presence of anti-PL7 antibodies due to the higher risk of ILD progression [17].

### 5.2. Progressive Fibrosing Phenotype in IIM-ILD

The concept of progressive fibrosing phenotype has recently gained significant value in the management of ILD patients, especially since the antifibrotic agent nintedanib was shown to slow the progression of fibrotic ILD other than Idiopathic Pulmonary Fibrosis (IPF) and its use was subsequently approved for the treatment of these patients [88]. Unfortunately, it is difficult to directly extrapolate information for the treatment of IIM patients from the INBUILD trial, as the proportion of IIM-ILD patients recruited is unclear. However, the concept is of great interest, as it has been demonstrated that about 30% of non-IPF patients with a fibrotic phenotype on HRCT share a similar prognosis to classic IPF patients despite the underlying condition [89]. Progressive fibrosing ILD is not limited to UIP patients but is potentially also connected to fibrotic NSIP and OP, being defined by specific criteria including a relative decline of Forced Vital Capacity (FVC) of 10% or a decline of FVC between 5–10% combined with worsening of the fibrotic extension in HRCT or respiratory symptoms within 24 months [88].

It is reported that about 25% of ASyS-ILD present a progressive fibrosing phenotype, mainly associated with the seropositivity for non-anti-Jo1 antibodies and the presence of reticular opacities on HRCT. This group could benefit from antifibrotic treatment in addition to the classic immunosuppressive drugs [90].

There are no available data on PPF in the context of ILD-DM. However, some authors reported on the administration of the antifibrotic agent pirfenidone to clinically amyopathic dermatomyositis patients, showing no effects in the acute phase but a trend for the reduction of mortality in the sub-acute phase [91].

Overall, these data highlight the importance of the correct recognition of ILD-IIM patients in order to avoid their misclassification as indeterminate or idiopathic ILD cases with the consequent risk of receiving incomplete treatment.

## 6. The Problem of Interstitial Pneumonia with Autoimmune Features

IPAF was proposed as a research classification to include ILD patients with some clinical and/or serological features of autoimmune disease that do not meet classification criteria for specific CTDs [66]. The concept resembles the idea of Undifferentiated CTD, used in rheumatology, and it could potentially include patients where the association between the ILD and autoimmune characteristics is entirely stochastic, but also patients where ILD is the first or even the sole clinical feature of a well-defined autoimmune disease [92]. Since its publication in 2015, the IPAF classification has also become common in clinical practice, with several concerns mainly regarding IIM. In fact, the criteria are divided into clinical, serological, and morphological domains, and classification requires at least one criterion from two different domains. In the clinical domain, IPAF criteria include Gottron Papules, which are considered pathognomonic for DM. Also, in the serological domain, the IPAF classification includes autoantibodies highly specific for IIM, such as anti-MDA and all of the ASA. The morphological domain includes all possible HRCT patterns of ILD, excluding the UIP pattern, which is deemed to be insufficiently associated with autoimmune conditions. The main difficulty is that ILD is not considered in the historical Bohan and Peter criteria for DM and is not mentioned even in the 2017 criteria for IIM. ASyS still lacks validated classification criteria (although they are expected to be available in early 2025). Therefore, ILD patients with positivity for ASA might be classified as ASyS or IPAF depending on the clinical setting (the first being a reasonable diagnosis in a rheumatology unit and the second in a respiratory setting), with possible implications on treatment. This is especially true for UIP-like patients with ASA positivity. This condition is likely sufficient for a rheumatologist to classify as ASyS (and therefore suggest immunosuppressive treatment) but not sufficient for a classification as IPAF. Consequently, in some respiratory settings, these patients may be classified as IPF and treated with antifibrotic drugs. Retrospective studies on IPAF have reported a surprisingly high prevalence of ASA positivity, whereas this number is almost absent in prospective studies [67,93]. As expected, IPAF patients with MSA positivity show a prognosis similar to established ILD-IIM, further suggesting that a true positivity for MSA in ILD patients could be enough to make a specific diagnosis [94]. Diagnosis is particularly important in patients with fibrotic ILD, as those treated with antifibrotic medications have an almost identical prognosis to IPF, while those treated with immunosuppressive therapy show benefits even in a fibrotic phenotype [95,96].

In light of this, ILD patients with pathognomonic clinical features and/or a true positivity for highly specific MSA should be considered to be affected by IIM and treated accordingly. The IPAF classification has the merit of highlighting the potential autoimmune pathway underlying ILD; however, it requires revision of its criteria in light of new knowledge, mainly regarding MSA.

## 7. Conclusions

The recognition of an underlying systemic autoimmune disease in ILD patients is crucial, opening the way to significant therapeutic opportunities. Immunosuppressive drugs can stabilize or even improve lung involvement, also potentially preventing the development of other typical signs of the disease. This is particularly important for IIM because ILD is potentially severe and commonly even the sole clinical manifestation associated with the autoimmune condition.

Screening for a possible underlying IIM should be proposed to ILD patients showing an NSIP and/or OP pattern, but it should not be limited to these because about 30% of ILD-IIM patients show a UIP or indeterminate pattern.

All patients with ILD should be clinically evaluated for the presence of proximal weakness, myalgia and typical IIM skin rashes, which are of great diagnostic value. Recognizing them may not be easy for physicians who are not properly trained; therefore, we suggest close collaboration between pulmonologists and rheumatologists working together in lung units or at least on a multidisciplinary team. Indeed, this collaboration could be of mutual benefit.

In addition, all patients with ILD should be evaluated for the presence of serological clues of possible underlying IIM. A significant increase in muscular enzymes such as transaminases, CPK, aldolase and myoglobin (1.5–2× the upper limit) could be meaningful, even if not associated with proximal weakness.

Finally, the autoimmune profile should be thoroughly assessed in all patients with particular attention to the testing for MSA/MAAs in the presence of ANA positivity with a nucleolar or cytoplasmic pattern, positivity for anti-Ro52kD and/or clinical features suggestive of IIM, even if classic autoimmunity tests (such as ANA and ENA) are negative. However, positivity for MSA/MAA should be carefully evaluated for possible false positivity, particularly in the presence of borderline positivity and lack of compelling clinical features. A possible approach is summarized in Figure 4.

## Figures and Tables

**Figure 1 diagnostics-15-00275-f001:**
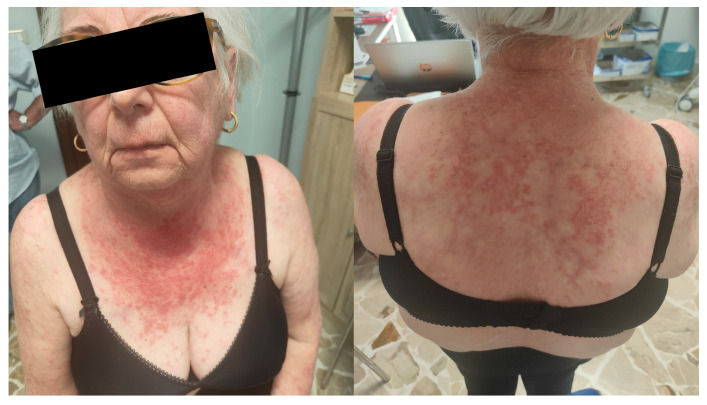
Shawl Sign present in the last year in a 68-year-old woman with DM.

**Figure 2 diagnostics-15-00275-f002:**
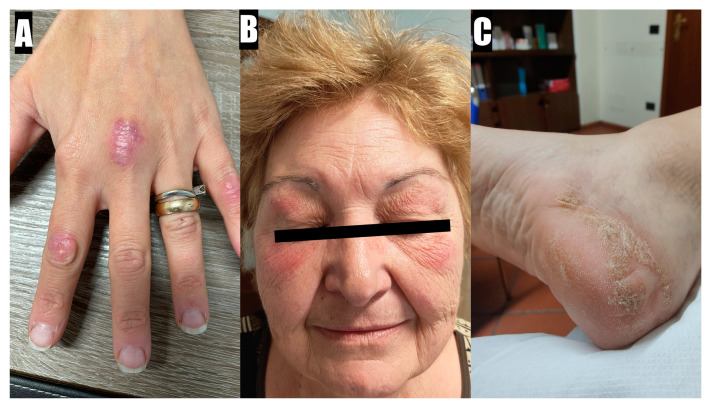
(**A**): Gottron Papules that first appeared a week before in a 43-year-old woman with DM; (**B**): Heliotropic Rash present in the last 6 months in a 64-year-old woman with DM; (**C**): Hiker’s Feet, undated, in a 53-year-old woman with ASyS.

**Figure 3 diagnostics-15-00275-f003:**
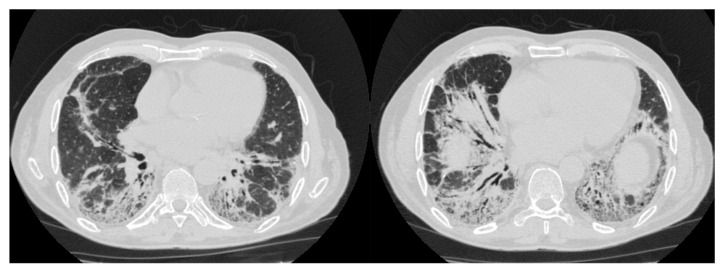
NSIP + OP pattern in a 57-year-old male at the time of diagnosis of DM.

**Figure 4 diagnostics-15-00275-f004:**
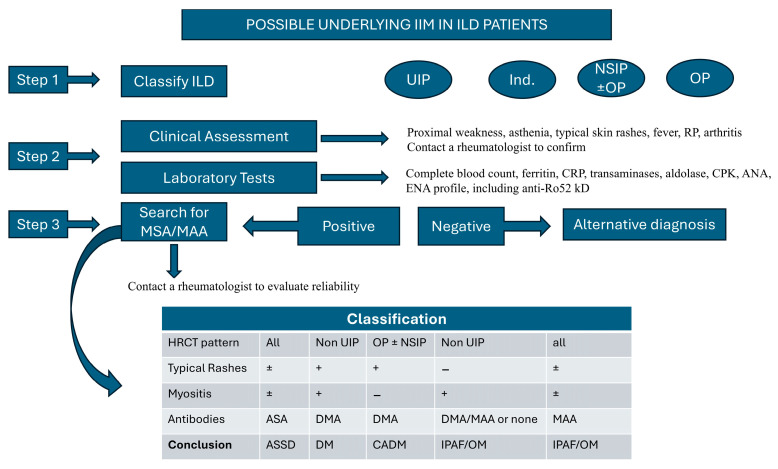
ANA: Antinuclear Antibody; ASA: Antisynthetase Antibodies; ASyS: Antisynthetase Syndrome; CADM: Clinically Amyopathic DM; CPK: Creatine Phosphokinase; DM: Dermatomyositis; DMA: DM Antibodies; ENA: Extractable Nuclear Antigen; IIM: Idiopathic Inflammatory Myopathy; IPAF: Interstitial Pneumonia with Autoimmune Features; HRCT: High-Resolution Computed Tomography; ILD: Interstitial Lung Disease; Ind: Indeterminate; MAA: Myositis Associated Antibodies; MSA: Myositis Associated Antibodies; NSIP: Nonspecific Interstitial Pneumonia; OM: Overlap Myositis; OP: Organizing Pneumonia; RP: Raynaud’s Phenomenon; UIP: Usual Interstitial Pneumonia. +: present; −: absent; ±: possible.

**Table 1 diagnostics-15-00275-t001:** Autoantibodies in Idiopathic inflammatory Myopathies.

Class	Subclass	Autoantibody	Target	ANA Staining	Condition	Clinical Picture
**MAA**	**MAA**	Anti-Pm/Scl [51,52]	Anti-Pm/scl exosome complex (100kD or 75 kD)	Nucleolar	Scleromyositis	ILD, DM skin rashes, myositis (100 kD), Scleroderma features (75 kD)
		Anti-Ku [53]	Ku (DNA binding protein)	Fine speckled	Scleromyositis, but also other CTD	SSc features, myositis
		Anti-Ro52kD [54]	TRIM21	Negative, fine speckled or cytoplasmic	ASyS, pSS but also other CTDs	Severe ILD, commonly associated with ASA
		Anti-U1RNP [55]	Subunit 1 of Ribonucleoprotein	Coarse speckled	MCTD	Mixed Connective Tissue Disease (ILD, arthritis, myositis)
**MSA**	**IBMA**	Anti-CN1A [56]	Cytosolic 5′-nucleotidase 1A	none	IBM	Weakness with mild increased CPK
	**IMNMA**	Anti-SRP [57]	Signal Recognition Particle	Cytoplasmic	IMNM	Myositis with little or no histologic inflammation
		Anti-HMGCR [57]	3Hydroxy-2-methilglutaryl-Coenzyme A reductase	Negative or cytoplasmic speckled	IMNM	Myositis with little or no histologic inflammation
	**DMA**	Anti-MJ/NXP2 [58]	NXP2	Multiple nuclear dots or speckled	DM	Classic DM, severe myositis at onset, possible association with cancer
		Anti-Tif1γ [59]	Transcription intermediary factor 1 γ	Fine speckled	DM (cancer-associated)	DM, mainly cancer-related. Possible hypo-amyopathic
		Anti-Mi2 [60]	Nucleosome remodeling deacetylase complex	Fine speckled	DM	Classic DM, mild
		Anti-MDA5 [61]	Melanoma Differentiation associated protein 5	Negative or faint cytoplasmic	DM (CADM)	Clinically amyopathic, potentially severe ILD and rashes
		Anti-SAE1 [62]	Small ubiquitin-like modifier activating enzyme	Speckled	DM	Classic DM
	**ASA**	Anti-Jo1 [34]	Hystidil-tRNA synthetase	Cytoplasmic fine speckled	ASyS	ASyS, classic triad ILD-inflammatory arthritis and myositis
		PL-7 [63]	Threonyl-tRNA synthetase	Cytoplasmic dense fine speckled	ASyS	ASyS, prevalence of ILD (potentially severe)
		PL-12 [63]	Alanyl-tRNA synthetase	Cytoplasmic dense fine speckled	ASyS	ASyS, prevalence of ILD (potentially severe)
		Anti-EJ [63]	Glycyl-tRNA synthetase	Cytoplasmic speckled	ASyS	ASyS, prevalence of ILD (potentially severe)
		Anti-OJ [64]	soleucyl-tRNA synthetase	Cytoplasmic speckled	ASyS	ASyS with prevalent ILD or even ILD alone
		Anti-KS [64]	Asparaginyl-tRNA synthetase	Cytoplasmic speckled	ASyS	ASyS with prevalent ILD or even ILD alone
		Anti-Zo [65]	Phenylalanyl-tRNA synthetase	Cytoplasmic speckled	ASyS	ILD commonly alone
		Anti-Ha [65]	Tyrosyl-tRNA synthetase	Cytoplasmic speckled	ASyS	ILD commonly alone

Legend: ANA: Antinuclear Antibody; ASA: Antisynthetase Antibodies; ASYS: Antisynthetase Syndrome; CADM: Clinically Amyopathic Dermatomyositis; CPK: Creatine Phosphokinase; CTD: Connective Tissue Disease; DM: dermatomyositis; DMA: Dermatomyositis Antibodies; IBM: Inclusion Body Myositis; IBMA: Inclusion Body Myositis Antibodies; ILD: Interstitial Lung Disease; IMNM: Immune Mediated Necrotizing myositis; IMNMA: Immune Mediated Necrotizing myositis Antibodies; MCTD: Mixed Connective Tissue Disease; MSA: Myositis Specific Antibodies; SSc: Systemic Sclerosis.

## Data Availability

Not applicable.

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
