# Peer review of "Recognition of Idiopathic Inflammatory Myopathies Underlying Interstitial Lung Diseases"

_diagnostics, 2025, doi:10.3390/diagnostics15030275_

Round 1

Reviewer 1 Report

Comments and Suggestions for Authors

 Recognition of Idiopathic Inflammatory Myopathies underly-2 ing Interstitial Lung Diseases

This manuscript covers an interesting topic. However, some points need to be addressed.

Page

Line

Manuscript

Comments

1

33

 Dermatomyositis and Anty-Synthetase Syndrome

No need for capitalization

 focusing on Dermatomyositis and Anty-Synthetase Syndrome

This point was not addressed in the title .

38

 Interstitial Lung Disease

No need for capitalization

English editing and grammar revision is mandatory.

44

 Connective Tissue Diseases (CTDs)

Again, capitalization without any need.

2

47

or it will develop during the

Present tense

2

48

and Idiopathic Inflammatory Myopathies (IIMs),

The authors did not follow the scientific rules of writing abbreviations.

The writing of the introduction is primitive and lack scientific rules.

The introduction should e divided into multiple paragraphs and each paragraph covers a specific idea and the introduction ends with the research gap and aim of the study.

2

70-71

Within the IIM spectrum, the two conditions most commonly associated with ILD 70 are DM and ASSD, therefore this review is mainly focused on these two entities. The clas-71 sification of DM is mainly based on the historical Bohan & Peter Criteria [10,11]

Flights of ideas.

Incoherence between the sentences in the same paragraph .and the same paragraph does not cover the same idea.

ASSD

This is not the scientific well-known abbreviation of this syndrome.

The authors did not determine if these figures related to their patients and if they obtained permission or not.

The footnotes of figures should include the description in detail regarding the patient age, gender, disease, its duration.

7

221

Interpretation and Reliability of Myositis Autoantibodies

The authors cover the disease in detail and with sophisticated details. However, the title focuses on lung diseases.

This makes the manuscript redundant and not focusing.

The manuscript that focuses mainly on lung disease did not contain any figure related to the lung affection in this population.

Author Response

Thank you for your insightful comments, our replies are the following

Row 1: thank you, fixed

Row 2: The Journal requires a short title. As explained in paragraph 2, the classification of IIM including IBM, IMNM, PM, DM, ASSD (of which the latter 3 associated with ILD) was recently questioned as being PM patients better classified among IBM, IMNM and ASSD. Therefore, the sole IIMs associated with ILD remain DM and ASSD, the topic of the manuscript. In our opinion, the title could be kept, being this explanation, beyond the clarification in paragraph 2, quite well known for the readers

Row 3: This is the first time in which we cited the term in the main text of the manuscript, therefore according to the instruction of the author we had to use the capitalization. If in the post process the editor deems that the capitalization is not necessary, it can easily be removed.

Row 4: thank you for your suggestion. The manuscript was now reviewed by a native English speaker.

Row 5: see row 3

Row 6: thank you, fixed

Row 7: Thank you for your suggestion. We changed the 4 IIMs with IIM, according to the acronym used by ACR and EULAR (see ref 12)

Row 8: Thank you for the interesting suggestion. Being the question to assess only one (the difficult in recognizing IIM underlying ILD), we deemed not useful to divide the introduction in multiple subsections. Anyway, we tried to clarify the introduction paragraph, hoping that you can find this new version acceptable.

Row 9: thank you again, we tried to better clarify this paragraph.

Row 10: Actually “ASSD” is the third favorite acronym for the definition of Anti-Synthetase Syndrome, mainly for Europe. An international consensus involving 136 of the major researchers on Anti-Synthetase Syndrome (including me) made an effort to uniformize the acronym for the definition of this entity. The results, obtained after the first draft of this manuscript, will be submitted as an abstract for EULAR congress 2025. Therefore, we can start to use the acronym “ASyS”, approved by the study group with 93.1% of votes.

Row 11: we stated in the “informed consent statement” that we obtained informed consent for publication by the patients. The informed consent was also submitted together with the manuscript

Row 12: Thank you, fixed

Row 13 and 14: The title of the manuscript is not focused on lung disease, but on the recognition of underlying IIM in ILD patients. For the focus of the manuscript a concise guide for the correct interpretation of the autoantibodies (basically not present in other manuscripts) is in our opinion more useful. We can add a new figure, with typical ILD associated with DM and ASyS, however, radiological features of ILD had a significantly lower diagnostic weight than autoantibodies (see ref 17).

Reviewer 2 Report

Comments and Suggestions for Authors

The review is well written and I do not seee any particular problemi with its publication.

Some minor changes in affiliation of author n 4 and 5.

I understand that the authors' aim is to describe the clinical, serological and radiological features associated with IIM-ILD. Possibly the respirologist or the physician devoted to respiratory medicine is the first objective of this paper. 

As a consequence after the introduction and classification sections, to the clinical feature paragraphs, I firstly suggest to include a section (3.1) with respiratory sings and or symptons, such as cough and exertional dyspnea. Again, as the pulmonologist would be the focus of the study, I suggest to include, when the doubt of ILD is raised, as 4.1 section the imaging paragraph (now section 5) and as subsequent sections other paragraphs including ematochemical examination including autoantibodies, and so on. Possibly a paragraph on respiratory function such as diffusion, spirometry, walking test is also useful.

Regarding the autoantibodies paragraph as conclusion, I suggest to search for an algorhytm, that at present is not fully clear. For instance, the phrase to lines 381-385 might be OK "the autoimmune profile should be thoroughly assessed in all patients with particular attention to the testing for MSA/MAAs in the presence of ANA positivity with nucleolar or cytoplasmic pattern, positivity for anti-Ro52kD and/or clinical features suggestive of IIM, even if classic autoimmunity tests (such as ANA and ENA) are negative. Please consider to insert before Figure 3

Are you sure that high transaminase levels suggest the execution of autoanatibodies profile?

Are you sure that the paragph 6 is OK. I suggest to shorten it and insert some phrases into the conclusion section

Author Response

The paragraph on respiratory signs, symptoms and PFTs was excluded only because these signs/symptoms are not useful to distinguish between IIM-ILD and any other causes of ILD. However, if you deem that it can be useful, we added some sentences in lines 292-304. Similar discussion can be made for the radiological pattern of ILD. Despite NSIP +/- OP is a common pattern of IIM-ILD, it doesn’t have a sufficient diagnostic value to distinguish IIM-ILD from other causes of ILD. For example, ILD associated with primary Sjogren’s Syndrome shows the same radiological pattern of IIM-ILD, but generally with a significantly better prognosis. So, we would keep the order of the paragraph.

Unfortunately, a diagnostic algorithm is not currently available. We tried to propose an algorithm in figure 3.

We are sure that transaminases can suggest the MSA/MAA profile in ILD patients. It is quite common and sustained by a lot of literature. These enzymes are not specific for muscle, but they have a value as a clue.

Paragraph 6 is in our opinion very important. As explained in this paragraph, a lot of IIM-ILD are generally classified by pulmonologists as IPAF for the difficult to recognize rheumatological items. After almost 10 years from the publication of these criteria, we know that the presence of a reliable MSA is quite sufficient to consider ILD as associated with IIM. 

Reviewer 3 Report

Comments and Suggestions for Authors

Thank you for giving me the opportunity to review this article. Giulia Morina et al. emphasized the clinical significance of recognizing idiopathic inflammatory myopathies (IIM) underlying interstitial lung disease (ILD). They comprehensively reviewed the clinical, serological, and radiological features of IIM and introduced an algorithm for screening IIM in patients with ILD. This review article is insightful and significant for clinical practice. I suggest including the circumstance of acute exacerbation of ILD (AE-ILD), which presents with diffuse ground-glass opacity (GGO) and organizing pneumonia (OP), and combining it with pneumothorax or subcutaneous emphysema. This is because MDA5-positive dermatomyositis (DM) and some subtypes of antisynthetase syndrome (ASSD) can initially present with this pattern. 

Author Response

Thank you for your comments

As you suggested, we included cited the possibility of pneumothorax and subcutaneous emphysema in line 337, adding a figure with a classic active NSIP+OP pattern